# But at What Cost? Healthcare Utilization of Canadian Carer-Employees

**DOI:** 10.3390/ijerph21121686

**Published:** 2024-12-18

**Authors:** Regina Ding, Linda Duxbury

**Affiliations:** Sprott School of Business, Carleton University, Ottawa, ON K1S 5B6, Canada; lindaduxbury@cunet.carleton.ca

**Keywords:** caregivers, caregiver burden, healthcare utilization, aging

## Abstract

Caregiving plays a crucial role in aging societies by supporting individuals with chronic illnesses, disabilities, or aging-related needs. The unpaid labour provided by caregivers diverts healthcare resources from the formal healthcare system; however, this incurs costs to the caregivers themselves in terms of declines in personal wellbeing. This study explores the relationship between caregiving and healthcare spending for two groups of caregivers: eldercare only and sandwiched. We found that physician visits were the most common resource used by caregivers, at a mean of 3.69 (SD = 4.01) visits over a 6-month period, excluding non-users of this service. This was followed by mental health services (M = 5.86, SD = 7.02), emergency room visits (M = 1.77, SD = 1.38), and hospital admissions (M = 3.61, SD = 8.53). There were no significant differences in healthcare utilization between eldercare and sandwiched caregivers. There were mixed results regarding characteristics associated with greater resource use; however, the hours of weekly caregiving were most consistently associated with greater healthcare utilization, indicating that healthcare use may increase with care burden.

## 1. Introduction

Global population aging is associated with increases in healthcare demands and with greater healthcare spending on costly conditions such as chronic and degenerative disease. This poses significant challenges to healthcare systems worldwide, including Canada, where a large portion of all healthcare services is publicly funded. Recently reported data showed Canada spent approximately 12.2% of its gross domestic product (GDP) or CAD 331 billion in 2022 on healthcare [1,2].

Amidst these challenges, caregiving—the unpaid provision of care services by friends or family—has emerged as a potential solution to reduce governmental healthcare spending. It has, in fact, been estimated that caregiving provides an estimated net benefit to the Canadian government of CAD 4.4 billion per year [3]. While caregiving has always been present to an extent in society, the reduction in publicly funded eldercare services has meant that more Canadians are engaged in care provision than ever before—either by choice or because they lack alternative options. Approximately 25% of all Canadians are currently providing care to an elderly dependent, with this number expected to increase as the population continues to age [4].

The caregiving role is associated with a number of adverse consequences for the caregiver themselves. Caregivers often experience higher levels of stress, depression, and poorer health compared to non-caregivers, which paradoxically often means that caregivers make more use of the healthcare system [5,6,7]. This consequence underscores the complexity of evaluating the net benefit of caregiving to the healthcare system and places caregivers as both sources of resource alleviation as well as high users of the healthcare system.

A critical gap in understanding this intricate dynamic between caregiving and cost savings emerges. The nuanced interplay between the benefits to society and the individual costs incurred by caregivers remains a relatively unexplored area, highlighting the need for studies to evaluate the net impact of unpaid caregiving on both caregivers themselves and the healthcare system overall. Considering these challenges, this study explores the healthcare utilization patterns of employed Canadian caregivers to determine if engaging in caregiving is a net cost-saving measure for the Canadian healthcare system. We sought to answer the following questions: (1) What are rates of health service utilization among employed caregivers in Canada? (2) What characteristics of employed Canadian caregivers (i.e., sandwich vs. eldercare) are associated with higher or lower utilization rates of the healthcare system? Our literature review investigated the impact of caregiving on healthcare utilization, government initiatives to support caregivers, and the health outcomes of caregivers. In our analysis, we used number of physician visits, inpatient days spent in hospital, number of emergency room visits, and mental health appointments as metrics for healthcare utilization. We then examined predictors of these metrics among employed Canadian caregivers using a zero-inflated Poisson generalized linear regression model. The model aimed to identify characteristics associated with higher or reduced healthcare use, providing insights into the factors influencing healthcare needs within the employed caregiver cohort. This model enabled a comprehensive understanding of the complex interplay between caregiver characteristics and healthcare utilization patterns, shedding light on who, among employed caregivers, are using the most resources. The data presented in this paper can be used by policymakers to evaluate the status quo and help make a case for change with respect to policies and programs to support employed caregivers.

## 2. Literature Review

### 2.1. Aging Populations Increase Healthcare Demands

Increases in human longevity as a result of advances in medicine and better living conditions mean that the contribution of non-communicable diseases to global disease burdens has increased in recent decades [8]. Older adults (65 years+) make greater use of healthcare resources due to the higher prevalence of chronic or degenerative disease in this group. Annual healthcare spending on older adults is substantive, at CAD 12,000 per adult over 65 years of age, compared to CAD 2700 for Canadians under 65 [9]. Projections indicate that aging societies will stress healthcare systems and healthcare budgets, with an estimated 88% increase in healthcare expenditures from 2019 to 2040 due to the increase in the number of adults over 65 years of age in the population [10].

While population aging exacerbates the strain placed on the Canadian healthcare system, the issue has been worsened by healthcare resource scarcity. The economic downturns of the early 1980s in Canada resulted in fiscal pressures to reduce federal government spending and implement budget cuts. This was achieved by reducing healthcare spending and resource allocation for hospital beds and outpatient facilities and by introducing wage freezes for healthcare workers [11]. In the 1990s, the federal government transferred responsibility for healthcare to provincial entities. This resulted in further cuts to health services as provinces struggled to fund a multiplicity of facilities [12]. From the 1980s to 1996, an estimated CAD 30 billion was trimmed from the federal healthcare budget [13].

These cost-cutting measures have had lasting effects on the delivery of healthcare in Canada. Hospital/specialist treatment wait times have increased from 9.3 weeks in 1993 to 27.4 weeks in 2022 [14]. The waitlist for a long-term care bed has approximately doubled in length since 2011 in Ontario, with 40,000 older adults waiting for a bed at any time [15]. The lack of resource allocation for long-term care or hospital beds mean that institutional care is relegated from publicly funded professionals to the community or expensive privately paid services. Workforce shortages for healthcare providers only further exacerbates issues of access [16].

### 2.2. Caregiving as a Means of Cost Saving

In recent years, the Canadian government has increased its reliance on informal caregivers with the goal of reducing healthcare costs. In Canada, 75% of all eldercare, including the provision of medical assistance, personal support, transportation, and financial and emotional caregiving, is now provided by family members [17]. This assistance (which is often several years in duration) enables care recipients to remain at home and reduces the use of institutional care. Informal caregiving has been found, therefore, to reduce the use of formal healthcare services, such as hospitalizations, nursing homes, and long-term care facilities by those 65 years or older [18,19].

The labour provided by caregivers is vital in bridging gaps in the healthcare system. By acting as liaisons between care recipients, healthcare providers, and community support services, caregivers ensure the continuation of care for their family members while optimizing resource utilization and reducing healthcare fragmentation. It has been estimated that informal caregivers save the Canadian government CAD 25 billion dollars annually through their unpaid labour [20]. Other research shows that caregiving is associated with better care recipient health outcomes, an association that is linked to enhanced advocacy as well as the personal services being provided by the caregiver [21,22].

### 2.3. Government Initiatives

The strategy of reducing healthcare spending by diverting to caregiving is supported by the Canadian government, with several federal and provincial supports/incentives provided to lower the barriers to the provision of care. For example, three federal programs offer permanent residency in Canada to foreign workers who come to Canada to provide care [23]. This program has resulted in increases in the number of workers in the homecare sector and facilitated the provision of in-home or personal caregiving services by homecare agencies—services that traditionally have been delivered by hospitals and other healthcare facilities. These programs have, therefore, reduced some of the strain on the public healthcare system. The Canadian federal government also offers several tax credits and/or employment insurance benefits (e.g., the Canada Caregiver Credit, the Compassionate Care Benefit, and two types of Family Caregiver benefits) as financial aids to reduce caregiving expenses [24]. In Canada, caregiving is also protected under both the Canadian Human Rights Act (CHRA) as well as the Canada Labour Code [25]. These legal protections safeguard caregivers from workplace discrimination related to their caregiving responsibilities.

While on the surface it would appear that Canadian caregivers receive a high degree of support when they take on this role, these governmental supports are sometimes criticized as over-restrictive in their eligibility requirements or inadequate to support the needs of caregivers. Previous research has found that despite legal protections, few caregivers use these resources (particularly workplace supports such as caregiving leave and flexibility), as they fear stigmatization in their careers [26]. Additionally, these resources were designed to help caregivers giving high-intensity care as well as those who have been engaged in caregiving for long periods of time. They do not assist citizens who wish to find alternatives to caregiving in situations where caregiving is not a feasible arrangement. Ultimately, while government initiatives assist caregivers in some capacity, it would appear that they are insufficient as resources to protect against adverse consequences that arise from caregiving responsibilities.

### 2.4. Caregiver Health

Statistics Canada (2020) estimates that there are currently 7.8 million caregivers in Canada (i.e., approximately one in four Canadians currently are engaged in caregiving) [27]. The high prevalence of caregiving in Canada and elsewhere is consistent with the fact that this is the preferred form of care arrangement for both caregivers and their care recipients [28]. For many families, caregiving is also a culturally significant activity, highlighting filial piety and empathy. While many want to (and feel it is their duty to) provide caregiving for a loved one, a number of factors, including increasing geographic mobility, the rise of dual-income families, delayed childrearing, and the increased participation of women in the labour force, have made caregiving discordant with contemporary lifestyles [29]. These frictions, combined with the lack of affordable alternative arrangements for care provision, mean that caregiving has become increasingly difficult even in the face of government supports.

While caregivers alleviate healthcare utilization and spending, the caregiving role itself is not without its own costs. Pearlin’s stress-process model of caregiving, for example, posits that the caregiver role produces chronic stress as a result of a complex interacting network of stressors (i.e., direct caregiving demands, changes in employment, etc.), an inability to access needed resources (access to social services, financial aid, etc.), and contextual factors (i.e., caregiver demographics, financial capital, etc.) [30]. In particular, the model emphasizes that caregiver outcomes such as poor mental and physical health are cumulative and arise from ongoing caregiving demands exceeding the availability of resources to caregivers. This model provides support for the idea that caregivers often experience poorer health outcomes because the demands associated with caregiving in conjunction with other role demands, including employment and childcare, outpace resource availability.

The empirical literature provides support for Pearlin’s model. Studies have found that caregivers have higher levels of stress, depression, and physical health challenges compared to non-caregiving peers [31,32]. They also make greater use of the healthcare system compared to non-caregivers [33]. Worsening caregiver health outcomes are known to be associated with cases of high care burden, which are common in situations with end-of-life care recipients or recipients with severe disease [34]. Indeed, Ornstein et al. (2015) created a model that illustrates how caregiver health-seeking behaviour is extrinsically linked to caregiving experiences such that high-intensity caregiving and the high burden of care leads to reduced preventative healthcare and increased urgent/emergency care use for caregivers [35].

An emergent subgroup of caregivers known as sandwiched caregivers has become a notable group of interest to both researchers and employers in recent years due to their compounding care roles. Sandwiched caregivers provide care to both aging family members and dependent children. They are known to manage considerable strain on their time, emotional energy, and financial resources, often leading to higher levels of stress and burnout compared to caregivers who focus on a single care recipient [36,37]. Unfortunately, little research has been carried out on this group, as most of the literature focuses on care provision for a single group of recipients (i.e., those who provide childcare or eldercare). Our paper addresses this issue by not only examining sandwiched caregiver healthcare resource use but also by comparing their use with those who provide eldercare only. As such, we contribute new insights into the complex challenges faced by this group.

It is known that increasing caregiver burden has direct repercussions on caregivers’ own health. As caregivers’ health declines, their own healthcare utilization tends to increase, creating a paradoxical relationship between caregiving, averted healthcare spending, and healthcare costs due to the diminished health of the caregivers themselves. The concept of a net benefit of caregiving to the healthcare system therefore becomes complex when considering caregivers as a vulnerable cohort in need of care themselves. This paper sheds light on this issue by providing data that speak to the costs to Canadian society of an over reliance on caregivers to support an aging population.

## 3. Method

### 3.1. Data

In 2018, a data collection inquiry was made to several large-sized (500+ employees) Canadian employers encompassing public-sector, not-for-profit, and private-sector workplaces. The goal of the study was to explore the relationships between work obligations, familial responsibilities, and caregiving duties. Ten organizations (evenly split between sectors) ultimately consented to partake in the study. These organizations used their internal email system to send a note to their entire employee base (employers in Canada do not record information on the caregiving responsibilities of their employees) explaining the goals of the study and asking interested employees to participate in a survey focusing on employed caregivers. A link to an electronic survey to their employees was also included in the note from the employer. A set of screening questions were included on the first page of the survey to ensure all respondents were full-time employees who actively undertook childcare and/or eldercare responsibilities. The ensuing analysis presented in this paper concentrates on a subset of respondents (n = 1674) specifically engaged in eldercare activities. As required by the ethics committee at our university, all survey respondents consented to the collection of their data, and responses were aggregated to ensure anonymity and hide identifying information.

Demographic information of the survey respondents is reported in Table 1. Missing values for each variable range from 0.8 to 2.2%. Overall, we found our sample to be slightly older than the average Canadian (49.7 years old), and it was skewed toward female (69.2%). This profile is consistent with the Canadian employed caregiver demographic profile [38]. Most survey respondents were urban (87.8%) residents of populous provinces (two-thirds of the respondents lived in Ontario, British Colombia, or Quebec) who earned between CAD 60,000 and CAD 99,999 per year. Just over half of our respondents (59%) were knowledge workers with at least one university degree (52.9%). Time spent in caregiving in one week by the people in our sample was substantive (14.2 h on average per week).

### 3.2. Measures

Our survey included several healthcare utilization metrics. Specifically, we asked respondents to think back over the last 6 months and indicate the number of physician visits (all specialties), days of inpatient hospitalization, emergency room visits, and mental health visits with a mental health professional (i.e., family physician, psychiatrist, or therapist) made during this period. All healthcare utilization metrics are discrete integer data. These metrics were selected for inclusion in this study because they encompass the majority of ways that Canadians interact with the healthcare system [41]. Comparative data were also available for each of these metrics [41]. Respondents also reported demographic information such as age, sex, income, job classification, childcare status, location, and weekly hours spent in eldercare caregiving. We also asked those respondents who provided care to a dependent 18 years of age or younger to tell us the weekly hours spent in childcare caregiving. Given that all respondents provided eldercare, the childcare variable was then used to identify respondents in the sandwiched subgroup. This term is used in reference to caregivers who are “sandwiched” between two types of care provision: eldercare and childcare. Sex (male/female), job classification (knowledge worker/non-knowledge worker), and location (urban/rural) were coded as dichotomous variables, with the former response as the reference category. The knowledge worker variable was selected for inclusion to differentiate between socio-economically advantaged white collar workers who are more able to negotiate flexibility in terms of when and where they work and other types of workers who typically do not have access to flexible work arrangements [37]. While participants provided an exact age in years, for income, participants indicated individual annual income, which was then divided into quartiles and coded 1–4.

### 3.3. Analysis: Healthcare Utilization Among Caregivers

This study is comprised of two analyses. The first part addresses our first research objective and provides a descriptive analysis of caregiver-employee healthcare utilization, with these interactions with the healthcare system monetized to determine medical costs specific to caregiver-employees. Summary statistics of each healthcare metric for the entire sample (n = 1674), for those in the eldercare group (n = 951), and for those in the sandwiched group (n = 723) are presented in Table 2. Given that usage of healthcare resources is right-skewed, with many respondents indicating null usage of healthcare resources in the prior 6 months, two means are reported, namely the total sample mean and the adjusted mean, which is calculated by removing zero responses and represents the mean usage of resources among users of the corresponding resource. This adjusted mean was used in further analyses.

Using the summary statistics of each healthcare utilization metric, we monetized the economic cost of healthcare resource utilization from the medical system’s perspective. Our study provides a conservative or partial estimate of healthcare costs, as it is based only on four metrics of healthcare resource utilization (physician visits, inpatient hospitalization days, emergency room visits, and mental health appointments) and does not take into account inflationary factors. Mean costs associated with each of the four-healthcare metrics were taken from the literature, as identified below. To increase generalizability, we used pre-COVID mean costs whenever possible.

The following estimates were used in this analysis. First, the mean cost to the Canadian healthcare system associated with a single physician visit (across all specialties) is estimated at CAD 73.45 per visit, assuming one service per visit [1]. Second, the costs associated with a single day of acute inpatient hospitalization is estimated to be CAD 1058.19. This figure was derived from the mean cost of a hospital admission at CAD 7803, divided by the average length of a typical stay at 7.2 days [1,42]. Third, CIHI (2020) estimates that a single ER visit costs approximately CAD 304; this amount includes overhead and allied health costs such as diagnostic imaging and administration costs but excludes physician compensation [43]. Finally, the mean cost for a mental health appointment uses the rate of a typical physician visit, as it was previously found that in Ontario that 80% of mental health visits are in the form of a family physician visit [44]. Unlike the other healthcare resources considered in this study, mental health resources in Canada are not fully funded by the public healthcare system. As a result, the projected costs represent a societal medical cost rather than the cost to the public healthcare system.

To extrapolate total medical costs, we assumed a total of 6.1 million caregiver-employees in Canada, based on most recent estimates from Statistics Canada [38]. Recognizing that sandwiched and eldercare caregivers may have different healthcare needs, we further divided the 6.1 million caregiver-employees into an eldercare group and sandwiched group, based on the assumption that 28% of caregivers (1.7 million) have both childcare and eldercare responsibilities [38]. As a simplification, it was assumed the remaining 72% of caregivers (4.4 million) provide eldercare only, as age-related caregiving is the most common form of care provision [38]. The costs associated with a single eldercare or sandwiched caregiver were obtained by multiplying the adjusted mean use of the resource with the mean cost per visit derived from the literature. This per capita amount was then multiplied by both the estimated number of eldercare or sandwiched caregivers in Canada as well as our sample percentage of respondents (see Table 3). While we recognize that not all caregivers will use each of the services considered, we feel that our methodology provides a reasonable estimate of the total societal cost to the healthcare system incurred by caregivers in Canada.

### 3.4. Analysis: Predictors of Healthcare Utilization

In the second part of this study, we modelled caregiver healthcare utilization over a 6-month period using several zero-inflated Poisson regressions to determine which subgroups or characteristics were associated with use of each healthcare resource. Zero-inflated Poisson regressions are useful for modelling count outcomes that have a large number of zero responses [45]. They are distinct from traditional Poisson regressions in that zero-inflated regressions contain two components: a Poisson component for predicting count or integer values and a logit component for predicting zeros or null use [46]. As a result, zero-inflated Poisson models are amenable to healthcare studies since they are capable of distinguishing between two processes: (1) the decision to utilize a resource (user vs. non-user) and (2) the frequency of resource use. Majo and van Soest (2011) previously demonstrated that zero-inflated Poisson models (with fixed effects) outperform Poisson and negative binomial models for healthcare utilization data, as they can extricate the effects of non-users against high-frequency vs. low-frequency users [47].

Four separate zero-inflated Poisson regression models were fit to represent the four metrics used in this study: physician visits, inpatient hospitalization, emergency room visits, and mental health visits/appointments. Coefficients in the Poisson regression component represent the log mean of the dependent variable when all other predictor variables are constant. The incident rate ratio (IRR) is given by exponentiating the Poisson coefficients, β_1_, and can be interpreted as follows: for every unit change in the predictor variable, there is an associated change in the dependent variable by a factor of e^β1^. An IRR over 1 indicates an increase in the rate of an event/outcome, whereas an IRR less than 1 indicates a reduced rate of an event. The logit model explains excess zeros or null resource use (users vs non-users), where coefficient β_2_ represents changes in the log odds scale for the dependent variable per unit change in the predictor variable. The odds ratio (OR) is similarly obtained by exponentiating the coefficients of the logit model, e^β2^. Similar to the IRR, OR over 1 indicates an increase in the likeliness of an event/outcome (increased odds), whereas an OR less than 1 indicates reduced likeliness of an event (decreased odds). It should be noted that while interpretation of IRR and OR numbers are similar, they differ in that IRR and OR are conveyed in different units. IRR is expressed as a relative ratio between two rates, while OR is expressed as a relative ratio of likeliness of an event occurring vs. not occurring.

Models were created using R version 4.2.2 using the “pscl” package, using maximum likelihood estimation. Sex, location, and job classification were included as dichotomous control variables in all models by way of multiple regression. In addition, income, age, and weekly care hours were included as continuous control variables. Caregiver status was a categorical variable that indicated whether a respondent was providing eldercare only or was a sandwiched caregiver. A log likelihood-ratio chi-square test compared our proposed model against the null model (intercept-only) to test the null hypothesis that the null model is a better fit for the data.

## 4. Results

### 4.1. Part 1: Healthcare Utilization

Physician visit(s) within the past 6 months were the most common healthcare resource used among our sample of employed caregivers (n = 1712), with 56.95% of the sample reporting utilization of this resource (Table 2). Of the respondents that visited a physician, the mean number of visits in a six-month period was 3.69 (SD = 4.01). Many caregivers also sought care from a mental health professional (i.e., 18.86% our survey respondents indicating they had used this service within the prior 6-month period at a mean of 5.86 (SD = 7.02) visits in this period. ER visits averaged 1.77 trips (SD = 1.38) for 16.82% of our sample. The least commonly used resource was inpatient hospital admissions, with 8.06% of the sample averaging 3.61 (SD = 8.53) days of admission.

There were no significant differences in the use of resources between eldercare and sandwiched caregivers for physician visits (*p* = 0.885), inpatient hospitalization (*p* = 0.569), ER visits (*p* = 0.505), or mental health appointments (*p* = 0.341). The estimated healthcare spending for an eldercare-only caregiver in a 6-month period was estimated to be CAD 666.43 across all four healthcare metrics, accounting for prevalence of use and frequency of use. The total estimated cost for eldercare caregivers in Canada, assuming 4.4 million eldercare-only caregivers in total, was approximately CAD 2.93 billion. For sandwiched caregivers, the estimated healthcare spending over a six-month period was CAD 609.53 per caregiver, and the estimated societal medical cost was CAD 1.04 billion across all 1.7 million sandwiched caregivers. Altogether, the total medical costs associated with caregiver healthcare utilization was found to be approximately CAD 3.97 billion within a 6-month period or approximately CAD 8 billion dollars per year.

### 4.2. Part 2: Modeling Healthcare Utilization

The zero-inflated Poisson regressions estimated significant predictors of resource use among demographic and socioeconomic variables. All four models yielded significant *p*-values for the chi-square test on log likelihoods, indicating that our proposed models fit better than their corresponding null model. Several significant associations were found across the four models, with the coefficients, standard error, significance incident rate ratios (IRR), and odds ratios (OR) reported in Table 4. For brevity, only significant associations are described in the section below. Significant associations are summarized in Table 5.

***Physician Visits:*** The number of physician visits was negatively predicted by caregiver age (IRR = 0.995, *p* = 0.047), where an increase in age by one year was significantly associated with a decrease in physician visits by 0.5% (i.e., decrease by a factor of 0.005 per year). Residing in a rural location (IRR = 1.168, *p* = 0.011) was associated with an increase in number of physician visits by 16.8% compared to urban residents. Weekly hours of care (IRR = 1.006, *p* = 1.89 × 10^−13^) were positively associated with number of physician visits, where every hour of weekly care provided increases the number of physician visits by 0.6%. In addition to the IRR, the odds of having no physician visits (null use) increases 30.3% when the participant is male compared to women (OR = 1.303, *p* = 0.023); this is the only significant association in the logit component. The remainder of the non-significant IRR and OR are reported in Table 5.

***Days Spent in Hospital:*** Inpatient days spent in a hospital was negatively associated with sandwiched caregiver status (IRR = 0.654, *p* = 2.61 × 10^−4^), living in a rural location (IRR = 0.368, *p* = 7.89 × 10^−16^), and weekly hours of care (IRR = 0.971, *p* = 3.51 × 10^−9^). For every one-hour increase in weekly caregiving hours, caregivers have a decrease in inpatient hospitalized days by approximately 2.9%. Conversely, knowledge workers had 2.477 times more days spent in the hospital compared to caregivers employed in non-knowledge positions (IRR = 2.477, *p* = 3.51 × 10^−9^).

Interestingly, this relationship is reversed in the logit model, where being a knowledge worker increases the odds of never being hospitalized by 76.3% compared to non-knowledge workers (OR = 1.763, *p* = 5.67 × 10^−3^). In other words, caregivers who are knowledge workers are less likely to be hospitalized overall, but those knowledge workers who end up hospitalized spend more days in the hospital than non-knowledge workers. Male caregivers were found to have 34.7% higher odds of using any amount of hospital inpatient days (OR = 0.653, *p* = 0.0426) compared to female caregivers. The logit model also revealed weekly hours of caregiving were associated with greater resource use, with every one-hour increase in weekly caregiving hours associated with a 2.2% decrease in the odds of never spending days in the hospital (OR = 0.978, *p* = 2.45 × 10^−3^).

***Emergency Room Visits:*** Income was negatively associated with number of ER visits (IRR = 0.776, *p* = 0.012), with caregivers that have higher income levels also having an associated 22.4% lower rate of ER visits compared to caregivers with lower income levels. Knowledge workers engaged in caregiving demonstrated a positive association (IRR =1.322, *p* = 0.044), reporting a 32.2% higher rate of ER visits compared to caregivers in other positions. Hours of weekly care exhibited a negative effect (IRR = 0.446, *p* = 0.050) such that for every one-hour increase in weekly caregiving hours, caregivers experience a 55.4% lower rate of ER visits.

Within the logit model, older caregivers were less likely to utilize the ER than younger caregivers. For each one-year increase in caregiver age, the likelihood of never using the ER increased by 3.8% (OR = 1.038, *p* = 3.04 × 10^−3^).

***Appointments with Mental Health Professionals:*** Caregivers with childcare responsibilities (also referred to as sandwiched caregivers) had a 9.8% lower rate of appointments with mental health professionals compared to caregivers without childcare responsibilities (IRR = 0.902, *p* = 0.045032). Male caregivers had 14.7% more appointments with mental health professionals compared to female caregivers (IRR = 1.147, *p* = 9.95 × 10^−13^). Age had a negative association with mental health appointments (IRR = 0.984, *p* = 4.09 × 10^−7^), with every one-year increase in age coupled with a 1.6% lower rate of appointments with mental health professionals. Income reduced the rate of mental health appointments by 8.2% per quartile (IRR = 0.918, *p* = 0.0234). Knowledge workers had 21.6% increased usage of mental health appointments compared to non-knowledge workers (IRR = 1.216, *p* = 1.29 × 10^−4^).

Age exhibited an OR of 1.02 (*p* = 0.00304), indicating that every one-year increase in age is associated with 2.4% higher odds of not seeking an appointment with mental health professionals. Weekly hours of care similarly had a negative association with never using mental health services, where every one-hour increase in weekly caregiving hours is accompanied by 0.8% lower odds of having no appointments with mental health professionals (OR = 0.992, *p* = 7.77 × 10^−3^).

## 5. Discussion

This study investigates the healthcare resource utilization of caregiver-employees. Specifically, our study was designed to help us do the following: (1) understand the typical use of Canada’s healthcare system by two types of caregiver-employees (eldercare vs. sandwiched), (2) identify the factors that contribute to higher or lower utilization of the healthcare system, and (3) quantify the healthcare costs associated with employees engaging in informal caregiving. We estimated that the total healthcare resource utilization for employed caregivers incurs a cost of CAD 3,968,032,917.50 over a 6-month period (i.e., CAD 666.43 per eldercare caregiver and CAD 609.54 per sandwiched caregiver). On average, over a 6-month period, caregivers in our sample would expect to have (unadjusted) 2.1 visits to a physician. In comparison, the general working-age population (18–64 years old) in Canada numbers 22.3 million, with only 28% visiting a primary care physician three times a year or more [48]. Among this working age cohort, there were approximately 0.36 visits per working-age Canadian in 2021–2022 [1,49]. In our research, we observed an average of 0.30 ER visits in only 6 months. We report a mean of 0.27 inpatient days spent admitted in a hospital and 1.126 appointments with a mental health professional. Unfortunately, national averages for each of these outcomes by Canadians who are in the working-age cohort is not available, and we are therefore not able to say if this level of use is lower or higher than that observed for employees who are not engaged in caregiving. While CIHI reports the average length of hospital stay at 7.2 days, this average is likely inflated by hospitalizations of older Canadians [1,49]. These comparisons, while not perfect, provide support for the idea that employed caregivers make greater use of the four different types of healthcare examined in this study than other Canadians of working age. To our knowledge, this is the first study in Canada estimating all four of these outcomes comprehensively in the caregiver population.

The extant literature presents conflicting views on the relationship between caregiving and healthcare utilization. Hopps et al. (2017) found that caregivers had more frequent outpatient visits and more disease comorbidities than non-caregivers [50]. Conversely, a national American study found that caregivers had reduced inpatient admission and medical expenditures compared to non-caregivers [51]. As we did not have a non-caregiver control group in this paper (the survey was designed with caregivers in mind), we are unable to remark on this observation. However, our findings revealed lower utilization rates of healthcare services compared with previous research from other countries. Specifically, our data suggest a mean monthly number of physician visits across all specialties of 0.35 and an average of 0.046 monthly emergency room visits per person. Schultz and Cook (2011) previously found U.S.-based caregivers (n = 583) had a mean of 0.77 primary physician visits per month and 0.032 ER visits [52]. Similarly, Rahman et al. (2019) established in their sample of U.S. dementia caregivers (n = 72) that mean general practitioner visits over a 6-month period averaged 4.3 (0.72 visits monthly), and ER visits averaged 1.7 (0.28 visits monthly) [33]. A Dutch study reported a mean of 1.4 general practitioner visits in a 4-week period for caregivers (n = 46) of cancer patients [53].

These disparities in healthcare use may reflect the challenges Canadians face with respect to accessing healthcare services in a timely fashion as well as the downstream ramifications on the type of service patients seek in the face of healthcare scarcity. Lengthy wait times for medical appointments in Canada mean that people may turn to emergency care and urgent care centers for timely treatment for non-serious conditions. The problems in timely healthcare access in Canada are well documented, with approximately 4.6 million Canadians (14.5% of the population) lacking access to a primary healthcare provider [54]). Specialist service wait times are at a historic high, at a median of 27.4 weeks to see a specialist [14]. This trend is also seen with mental health resource use, where, in our sample, the mean use of mental health appointments (1.126 or 0.188 monthly) was lower compared to the reported 0.368 appointments per month per caregiver found by Schultz and Cook [52] (2011). However, comparisons of mental health resource use across different contexts are limited, as mental health services are not fully publicly funded in Canada. Currently, the literature on caregiver days spent in the hospital is insufficient.

It should be noted that these discrepancies in healthcare utilization may also be due to differences in sample size (in general, the studies we reference above had smaller sample sizes than ours) and composition (our sample included only employed caregivers, which is not necessarily the case in the studies reported in the literature). It is, therefore, conceivable that observed differences in healthcare utilization arise from differences in sample characteristics. It is also important to note that the costs presented above do not fully capture total societal costs of caregiving, as we did not capture opportunity costs incurred by caregivers because of labour participation adjustments.

The zero-inflated Poisson regression identified several significant factors that influenced caregiver use of healthcare resources. Our results found that the variable of weekly hours of care was one of the most consistent variables that predicted healthcare usage. We note that caregivers with higher weekly caregiving hours (1) were more likely to have a greater number of physician visits and ER visits and (2) spent fewer nights in the hospital. These results imply that caregivers who devote more time to caregiving may experience poorer health (both physical and emotional), requiring them to seek medical attention more frequently. We speculate that some caregivers visit the ER when long wait times make it difficult for them to seek physician care. These findings are consistent with those reported by Bremer et al. (2015) [55]. Their multi-national analysis also reported a strong positive association was found between daily caregiving hours and caregiver’s reported healthcare utilization. Similarly, Ornstein et al. (2015) demonstrated similar results as our findings, where caregivers with a high care burden utilized more costly healthcare services, such as urgent care, rather than engaging in preventative care services [35].

Why are weekly care hours negatively associated with inpatient hospital admission? We suggest that these findings may be explained by the positive association between weekly hours of care and physician appointments. Specifically, we suspect that physicians are more likely to intervene in a timely fashion when the caregiver reports emerging health concerns. This, in turn, reduces the likelihood that the caregiver will develop severe health problems that require hospitalization. This interpretation also elucidates the negative association between weekly hours of care and the absence of resource utilization for both inpatient hospital days and mental health appointments. Due to early detection from frequent physician visits, caregivers who do become admitted to the hospital or seek mental health appointments represent caregivers with more serious health issues.

In summary, the relationship between weekly caregiving hours and healthcare utilization appears to be dynamic and complex. Caregivers with higher weekly hours of care may have more frequent use of routine healthcare services but reduced utilization of intensive and costly services such as hospitalization days. These findings emphasize the importance of access to routine care for caregivers as a cost-saving measure that averts more serious health conditions requiring expensive healthcare resources.

The relationship between knowledge work and healthcare utilization can also be considered multifaceted. The data show that the caregivers working as knowledge workers in our sample tended to be more socio-economically advantaged (higher education, higher incomes, etc.). This group of caregivers were more likely to have more days in the hospital, more frequent visits to the ER, and more appointments with mental health professionals. At the same time, being a knowledge worker was associated with greater odds of never spending days in the hospital. These findings are consistent with occupational health research showing that knowledge employees have higher incomes, fewer health risks, and greater access to high-quality healthcare, allowing them to handle health issues more promptly and proactively [56,57,58]. The higher earnings of knowledge workers have also been associated with a better work–life balance, meaning that if disease or injury does occur, they have the appropriate resources to stay in the hospital longer or seek care from the ER, physicians, or mental health professionals more frequently [59,60].

Interestingly, older employed caregivers made fewer physician visits and mental health appointments and were more likely to never use the ER and mental health services. Again, we can only speculate as to why older employed caregivers make less use of the healthcare system. One possible interpretation is that that older caregivers may be more experienced in resilience techniques and have greater familiarity with health-promotion behaviours via caregiving for their recipient, enabling them to effectively manage their own health and well-being while requiring fewer healthcare services [61]. This interpretation is consistent with a study performed by Burns et al. (2010), who found that younger caregivers reported greater difficulty providing care and were more likely to need bereavement leave than older caregivers [62]. Alternatively, older caregivers are more likely to be providing end-of-life care to an aging relative, which is often time-intensive. It may be possible that caregivers in these high-demand situations neglect or delay seeking appointments/services for their own health as they prioritize their care recipient [63].

Unexpectedly, we observed that male caregivers were more likely to have frequent mental health appointments compared to female caregivers. This contrasts with the literature, where men have been reported to be less likely to seek mental health resources [64]. We turned to research focusing on how male caregivers conceptualize and respond to caregiver burden and stress to help us understand these results. Research has found that male caregivers not only have greater commitment to work responsibilities but also are more likely to perceive there is a stigma attached to caregiving, which is viewed as a traditionally feminine role [65,66]). However, in recent years, there have been intensive efforts and numerous interventions dedicated to enhancing mental health literacy in men [67,68]. Our findings suggest that these interventions are bearing fruit, with male caregivers becoming more open to using mental health services. Alternatively, it may be that the need to engage as a caregiver is more stressful for men, who have not been socialized into this role.

In our study, we note that male caregivers also had reduced odds of not seeing a physician within the last 6 months compared to women. Our findings suggest that this strategy comes at a cost, as men are also more likely than women to be hospitalized—perhaps because they have not attended to their health issues in a timely fashion. This finding is supported by Lin et al. (2024), who previously found that male working caregivers had a higher frequency of heart disease than female working caregivers [69]. Like the relationship observed in caregivers with high weekly care hours, the lack of physician visits may lead to missed early detection of disease/injury, leading to worse health and requiring hospitalization.

One of the key factors that emerged from our analysis was the influence of income on healthcare use among caregivers. We found that caregivers with lower incomes had higher utilization of ER and mental health appointments. These findings may be attributed to the fact that caregivers with lower incomes may lack access to routine or primary care, which requires them to use acute and emergency care when they are ill. This finding suggests that financial resources may improve overall health or access to routine/preventative care, a conclusion aligned with studies carried out by Andrén and Elmståhl (2007), Fast et al. (2001), Sörensen and Pinquart (2005), Limpawattana et al. (2013), and Jeong et al. (2015) [70,71,72,73,74].

The association between healthcare utilization and rural residence was mixed. Our analysis suggests that caregivers residing in rural areas had more physician visits compared to their urban counterparts. However, the opposite was true for inpatient hospitalization days, where rural residents had fewer days spent in the hospital. The utilization of physician visits is contrary to what is suggested in most of the literature, where rural regions often have shortages of healthcare providers and spatial inaccessibility [75]. However, Haggerty et al. (2014) noted a similar finding to our study, with rural residents reporting greater organizational accommodation to their healthcare needs than urban residents, with participants citing their successful use of social connections to access healthcare rather than institutional pathways [76]. This same study found that rural residents also typically invoke information gathering and telemedicine earlier in their health-seeking trajectory than urban residents [76]. The lower rates of hospitalization days of rural residents can be attributed to factors such as limited access to healthcare facilities and transportation challenges, given that hospitals are likely located in urban areas. Regardless, addressing these geographic disparities in healthcare access and infrastructure is crucial to ensure that rural caregivers receive adequate and timely healthcare services.

One of the more intriguing trends captured in our analysis related to the fact that there were very few differences in healthcare utilization related to whether the caregiver also engaged in childcare. The only significant association noted was the negative relationship between childcare responsibilities and inpatient hospital days, where sandwiched caregivers had significantly fewer days spent in the hospital than caregivers who only provide eldercare. This contrasts with the literature on sandwiched caregivers, where sandwiched caregivers are known to experience higher strain due to the handling of multiple conflicting responsibilities [77].

The responsibilities associated with caring for children and aging parents simultaneously may lead to increased awareness and proactive management of health conditions, resulting in fewer hospital admissions. Alternatively, having children at home may buffer the negative relationship between caregiving and mental and physical health, and/or sandwiched caregivers with children at home may advocate for earlier discharge from the hospital to provide care for their children [78]. Another interpretation is that sandwiched caregivers may be younger than eldercare only caregivers and have overall fewer health conditions [79].

## 6. Conclusions

While caregiving is often regarded as a cost-saving mechanism for the federal healthcare system, our findings emphasize that it comes with its own costs. This study underscores the substantial healthcare spending associated with employed caregivers in Canada and highlights the factors contributing to this expenditure. We estimated that caregiver healthcare utilization costs the Canadian healthcare system CAD 3 billion over a 6-month period. Our analysis also showed multiple significant positive associations between hours spent caregiving in a week and use of the healthcare system, providing support for the notion that the inherent demands of the caregiving role increase the use of the healthcare system and thereby cost Canadian society. With that said, we also note that the utilization of physician appointments by Canadian caregivers appears lower than that of caregivers in other countries, suggesting the impact of low accessibility and long wait times, particularly as rates of ER visits were nearly double in our sample compared to other international studies. We do, however, encourage caution with respect to the interpretation of these utilization rates, as differences in healthcare systems, research samples, and national context can also lead to these discrepancies. Regardless, the deficits in the Canadian healthcare system are well known. Given that caregiving was originally commodified decades ago in lieu of abundant publicly funded healthcare services for eldercare, the current state of healthcare in Canada speaks to the continued decline in healthcare resource allocation directed to eldercare. Our findings show that the Canadian healthcare system is highly reliant on eldercare provided by informal caregivers, without whom the already strained healthcare system would irrefutably become overwhelmed by the need to take on the costly disease burden of chronic and age-related health needs. Our findings highlight the need to consider caregivers as an essential and distinct cohort in terms of healthcare planning and resource allocation.

Finally, our study considers the diversity and complex interplay of factors that influence healthcare utilization within our sample of caregivers, such as childcare responsibilities, age, sex, location of residence, income, and knowledge worker status. The multiplicity of relationships we observed in our model emphasizes the importance of recognizing the unique circumstances and needs of caregivers based on their specific characteristics and backgrounds.

## 7. Limitations and Directions for Future Research

Our study is limited in that we did not have access to a meaningful non-caregiver control group. As such, we are unable to determine the general population baseline healthcare utilization and differentiate caregiving-specific costs and utilization. We were also unable to assess the extent to which interactions with the healthcare system are specifically related to caregiving burden or the caregiver’s personal health issues, given that our survey did not probe personal health information.

We also note that we extrapolated the overall caregiver population when calculating the medical cost of caregiving, assuming that the general caregiver population would follow the same pattern of healthcare utilization as our sample—an assumption that is hard to validate given the sparsity of research on the healthcare utilization of employed caregivers. Finally, cost estimates of each healthcare services were borrowed from the literature and reflect national means, while our respondents reside in various provinces across Canada. Cost per physician visit in particular represent an aggregate mean across all specialties and is an oversimplification of the true cost of healthcare use.

The impact of caregiving on caregiver health is well established. However, our study is novel in that we quantify the costs of caregiving to the formal healthcare system in Canada. Jacobs et al. (2013) previously conducted an economic evaluation of caregiving costs and benefits to the Canadian government; however, they did not quantify changes in caregiver healthcare utilization as a result of their caregiving [3]. By exploring caregiver personal health and interactions with the healthcare, our study contributes to the literature on the system costs of caregiving from a medical perspective, as previous economic evaluations have been lacking in this element.

Future studies should explore comparisons between caregivers and non-caregivers as well as employed caregivers and those who are not in the labour force and explore the healthcare needs of these different types of caregivers as well as the accessibility of their healthcare resources. Additionally, given our findings, a number of suitable follow-up studies are possible. The data used in this paper were collected pre-COVID, (i.e., 2018) and the comparison data used in the paper are (by necessity) also somewhat dated. It would be beneficial to explore how inflation and other related economic factors have impacted the costs of caregiving estimated in this paper. We would also encourage scholars to engage in research to help policymakers better understand the factors that may moderate the relationship between participation in the caregiver role and the caregiver’s use of the healthcare system, such as the number of adults in the home, hours spent in work per week (full-time work versus PT work), use of flexible work arrangements, and the type(s) of impairment experienced by the care recipient (e.g., dementia, health condition, etc.). We can also see the need for research that explores the extent to which the effect of weekly hours of care on healthcare utilization measures is same or different for the “eldercare only” group and the “sandwiched” group.

Overall, our use of the zero-inflated Poisson regression also illustrated different patterns of healthcare use in the count model and logit model, highlighting that the decision to use a healthcare resource is predicated by a two-step process. This aligns with suggestions by Mao and van Soest (2011) [47].

It is also imperative to our research context to acknowledge the systemic challenges of the Canadian healthcare system, which exacerbates not only caregiver burden but the caregiver’s own health outcomes. Insufficient accessibility to healthcare not only impacts caregivers’ health and well-being but also hinders their ability to provide optimal care to their loved ones. Interventions aimed at addressing access to and delivery of healthcare resources may, therefore, benefit both the care recipient as well as the caregivers. By supporting caregivers in protecting their own health, policymakers and healthcare providers can not only enhance the well-being of an important cohort but also mitigate the broader societal, medical, and economic impacts of population aging in Canada.

## Figures and Tables

**Table 1 ijerph-21-01686-t001:** Sociodemographic composition of the sample.

Demographic Variable	Eldercare Only (N = 951)	Sandwiched ^1^ (N = 723)	Overall (N = 1674)
**Age**			
Mean (SD)	52.3 (7.84)	46.2 (7.13)	49.7 (8.12)
**Location Of Residence**			
Rural	117 (12.3%)	88 (12.2%)	205 (12.2%)
Urban	834 (87.7%)	635 (87.8%)	1469 (87.8%)
**Province**			
Ontario	393 (41.3%)	251 (34.7%)	644 (38.5%)
British Columbia	166 (17.5%)	158 (21.9%)	324 (19.4%)
Quebec	95 (10.0%)	79 (10.9%)	174 (10.4%)
Prairies (Albt., Man., Sask.)	163 (17.1%)	125 (17.3%)	288 (17.2%)
Maritimes (nb, ns, nfld, pei)	84 (8.8%)	77 (10.7%)	161 (9.6%)
Northwest Territories/Yukon	8 (0.8%)	0 (0%)	12 (0.7%)
**SEX**			
Female	710 (74.7%)	445 (61.5%)	1155 (69.0%)
Male	241 (25.3%)	278 (38.5%)	519 (31.0%)
**Highest Attained Education**			
High School Or Less	220 (23.1%)	129 (17.8%)	349 (20.8%)
College Diploma	255 (26.8%)	180 (24.9%)	435 (26.0%)
University Degree	318 (33.4%)	294 (40.7%)	612 (36.6%)
Post Graduate Degree	156 (16.4%)	117 (16.2%)	273 (16.3%)
**Personal Income**			
**Less than CAD 59,999**	237 (24.9%)	117 (16.2%)	354 (21.1%)
CAD 60,000 to CAD 99,999	499 (52.5%)	424 (58.6%)	923 (55.1%)
CAD 100,000 to CAD 149,999	177 (18.6%)	154 (21.3%)	331 (19.8%)
More than CAD 150,000	16 (1.7%)	15 (2.1%)	31 (1.9%)
**Knowledge Worker ^2^**			
No	362 (38.1%)	325 (45.0%)	687 (41.0%)
Yes	589 (61.9%)	398 (55.0%)	987 (59.0%)
**Weekly Hours Of Eldercare**			
Mean (SD)	16.0 (20.4)	11.3 (15.6)	14.0 (18.6)

^1^ Sandwiched caregivers refer to a growing sub-group of caregivers who provide childcare alongside eldercare; they have been a group of recent interest in the caregiving literature, as they are known for their unique role strains [39,40]. ^2^ Knowledge worker denotes a class of working professional that primarily uses knowledge as their primary skill (e.g., lawyers, physicians, etc.).

**Table 2 ijerph-21-01686-t002:** Healthcare utilization by caregivers in the past six months.

Healthcare Utilization Among Total Sample (n = 1674) in Past Six Months					
Healthcare Utilization Metric	% of Total Sample with Reported Usage of Service	Mean Usage	Std. Dev	Adjusted Mean Usage	Adjust Std. Dev
Physician visits	56.95%	2.1	3.54	3.69	4.02
Inpatient days spent at hospital	8.06%	0.27	2.52	3.61	8.53
Emergency room visits	16.82%	0.3	0.87	1.77	1.38
Mental health professional visits	19.86%	1.126	3.84	5.86	7.02
**Healthcare Utilization Among Eldercare Sample (n = 951) in Past Six Months**					
**Healthcare Utilization Metric**	**% of Eldercare Sample with Reported Usage of Service**	**Mean Usage**	**Std. Dev**	**Adjusted Mean Usage**	**Adjust Std. Dev**
Physician visits	57.42%	2.11	3.54	3.674	4.01
Inpatient days spent at hospital	7.68%	0.302	3.1	4.16	10.88
Emergency room visits	16.54%	0.307	0.965	1.88	1.66
Mental health professional visits	18.25%	1.05	4.01	5.87	7.9
**Healthcare Utilization Among Sandwiched Sample (n = 723) in Past Six Months**					
**Healthcare Utilization Metric**	**% of Sandwiched Sample with Reported Usage of Service**	**Mean Usage**	**Std. Dev**	**Adjusted Mean Usage**	**Adjust Std. Dev**
Physician visits	56.39%	2.09	3.53	3.7	4.02
Inpatient days spent at hospital	8.52%	0.236	1.55	3	4.74
Emergency room visits	17.16%	0.28	0.73	1.64	0.94
Mental health professional visits	21.81%	1.22	3.63	5.85	6

Note: The eldercare group (n = 951) includes respondents who are providing age-related care who do not have any childcare responsibilities (dependents under the age of 18 years). The sandwiched group (n = 723) includes respondents who spend time each week providing childcare as well as eldercare.

**Table 3 ijerph-21-01686-t003:** Estimated cost of healthcare resource use by caregiver employees within a 6-month period.

Service	Cost per Case/Visit (CAD)	Cost per Eldercare Caregiver ^1^ (CAD)	Cost per Sandwiched Caregiver (CAD)	Total Cost for Eldercare Caregivers ^2^ (CAD)	Total Cost Sandwich Caregivers (CAD)	Total Societal Cost All Caregivers (CAD)
Physician visit	73.45	154.94	153.24	680,512,282.76	261,734,592.18	942,246,874.94
Inpatient hospitalization (per day)	1058.19	338.26	270.35	1,485,641,576.09	461,760,554.84	1,947,402,130.93
ER visit	304.00	94.54	85.56	415,227,273.43	146,135,333.99	561,362,607.41
Mental Health Appointment	73.45	78.68	100.38	345,580,120.97	171,441,183.24	517,021,304.21
Total		666.43	609.53	2,926,961,253.25	1,041,071,664.24	3,968,032,917.50

^1^ Cost per eldercare/sandwiched caregiver calculated by multiplying cost per visit with adjusted mean usage of service and percentage of sample that reported usage of service from Table 2. ^2^ Total cost for eldercare/sandwiched group calculated from cost per eldercare/sandwiched caregiver multiplied by number of eldercare/sandwiched caregivers in Canada (assuming 6.1 million caregiver employees in Canada, with 28% as sandwiched and the remaining as eldercare-only caregivers).

**Table 4 ijerph-21-01686-t004:** Zero-inflated Poisson coefficients, significance, and odds rations/incident rate ratios.

		Count Poisson Regression				Logit Zero-Inflation Model			
		Coefficient	Std. Error	*p*-Value	IRR	Coefficient	Std. Error	*p*-Value	OR
Physician visits	Sandwiched	0.0243036	0.0393303	0.5366	1.024601	−0.01035	0.114807	0.9282	0.989705
	Sex (Male)	0.0266228	0.0412428	0.5186	1.02698	0.264569	0.116751	0.0234	1.302869
	Age	−0.0045877	0.0023085	0.0469	0.995423	−0.00766	0.006913	0.268	0.992371
	Rural	0.1554268	0.0609085	0.0107	1.168156	−0.15383	0.160824	0.3388	0.857419
	Income	−0.0074138	0.0275516	0.7879	0.992614	0.098445	0.079701	0.2168	1.103454
	Knowledge Worker	−0.0360957	0.0380053	0.3422	0.964548	0.002205	0.110304	0.9841	1.002207
	Weekly hours of care	0.0060507	0.0008225	1.89 × 10^−13^	1.006069	0.004356	0.002794	0.119	1.004366
Inpatient hospitalization	Sandwiched	−0.424363	0.116222	0.000261	0.654186	−0.1765	0.2136	0.408621	0.838196
	Sex (Male)	0.11827	0.105472	0.262146	1.125548	−0.42656	0.210395	0.04262	0.652752
	Age	−0.006129	0.006495	0.345359	0.99389	−0.00272	0.012789	0.831555	0.997284
	Rural	−0.999644	0.124088	7.89 × 10^−16^	0.36801	−0.27274	0.300498	0.364071	0.761289
	Income	−0.130796	0.071773	0.068402	0.877397	0.18176	0.150694	0.227758	1.199326
	Knowledge Worker	0.907307	0.121628	8.67× 10^−14^	2.477641	0.567242	0.205073	0.005674	1.763397
	Weekly hours of care	−0.029892	0.005061	3.51 × 10^−9^	0.97055	−0.02225	0.007345	0.002451	0.977994
ER visits	Sandwiched	−0.048039	0.154645	0.7561	0.953097	0.062116	0.198533	0.75438	1.064086
	Sex (Male)	−0.089614	0.150634	0.5519	0.914284	−0.30629	0.199998	0.12565	0.736172
	Age	0.010535	0.009834	0.284	1.010591	0.037339	0.012599	0.00304	1.038045
	Rural	−0.176952	0.184351	0.3371	0.83782	−0.00427	0.240871	0.98585	0.995737
	Income	−0.253679	0.101231	0.0122	0.775941	0.175103	0.135108	0.19497	1.191369
	Knowledge Worker	0.27902	0.138709	0.0443	1.321834	0.163304	0.185379	0.37836	1.177395
	Weekly hours of care	0.004454	0.002268	0.0495	1.004464	−0.00346	0.003765	0.35826	0.996547
		**Count Poisson Regression**				**Logit Zero-Inflation Model**			
		**Coefficient**	**Std. Error**	** *p* ** **-** **V** **alue**	**IRR**	**Coefficient**	**Std. Error**	** *p* ** **-Value**	**OR**
Mental health appointments	Sandwiched	−0.10329	0.051533	0.045032	0.901865	−0.14018	0.136965	0.3061	0.869205
	Sex (Male)	0.137107	0.053193	0.009951	1.146951	0.126059	0.143648	0.38019	1.134349
	Age	−0.016224	0.003203	4.09× 10^−7^	0.983907	0.024176	0.008158	0.00304	1.024471
	Rural	−0.032981	0.066677	0.620853	0.967557	0.215752	0.182311	0.23664	1.240795
	Income	−0.085395	0.037792	0.023847	0.91815	−0.05607	0.095649	0.55776	0.945477
	Knowledge Worker	0.1954	0.051045	0.000129	1.215797	0.045902	0.132168	0.72837	1.046972
	Weekly hours of care	0.001867	0.001013	0.06519	1.001869	−0.00816	0.003063	0.00777	0.991878

**Table 5 ijerph-21-01686-t005:** Summary table of significant associations.

		Frequency of Use		
	Use	Higher Use	Lower Use	Non-Use
Physician Visits		Rural location; weekly hours of care	Age	Male
Inpatient hospitalization (days spent in hospital)	Male; weekly hours of care	Knowledge worker	Sandwiched caregiver; rural location; weekly hours of care	Knowledge worker
ER visits		Knowledge worker; hours of weekly care	Income	Age
Mental health appointments	Hours of weekly care	Male; knowledge worker	Sandwiched caregiver; age; income	Age

## Data Availability

The data are not available for sharing due to privacy restrictions concerning organizational data.

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
