# Peer review of "But at What Cost? Healthcare Utilization of Canadian Carer-Employees"

_ijerph, 2024, doi:10.3390/ijerph21121686_

Round 1

Reviewer 1 Report

Comments and Suggestions for Authors

Overall Impression

The authors investigate a significant issue by examining healthcare utlization by caregivers in Canada and highlights the factors associated with utilization using a sample collected from 10 large sized organizations (500+ employees) in 2016. To this end, the authors include two groups of caregivers - "eldercare only" and "sandwich" group (those who provide both eldercare and childcare). This is a valuable area of research that has gained a growing interest in several countries due to aging population and increasing demands on healthcare, thus having an impact on healthcare cost, access, and quality. The paper is generally well-written and well-organized, easy to understand, and clear. I would like to offer some suggestions for improvement to strengthen the paper.

Suggestions:

While the premise of the study is sound, there are a few areas in need of written clarity and additional analyses.

Additional analyses:

1.      The study compares two groups of caregivers - "elder care only" and "sandwiched". The study is unable to compare the caregivers to individuals who do not provide any caregiving services. In the data section (line 178), the authors do mention that the study concentrates on the subset of respondents who engage in eldercare responsibilities. It is possible that the dataset does not contain a control group - no caregiving for the purpose of comparison. If this is the case, please provide in the footnote as to why a control group was not included. However, if the dataset does include a group of individuals that provide no caregiving services, then including them in the analyses can strengthen the implications of the study.

2.      The study focusses on showing the effect of caregiving on health care  utilization. The following variables have not been included in the analysis:

a.      underlying health status of the caregiver: If the caregiver has an underlying health condition (unrelated to caregiving), which can affect the healthcare utilization.

b.      relationship between caregiver and elderly person that needs care: Studies have shown that most elder caregivers are spouse of the individual, who themselves may have certain health condition.

c.      family size: Specifically, number of adult members in the household. Larger families may be able to share the caregiving responsibilities. In Table 1, we see that average weekly hours of eldercare is lower for sandwiched group. This could be because sandwiched group could have larger family size and thus responsibilities are shared among other members in the household.

d.      change in job: Prior studies have documented individuals that provide caregiving change their jobs - full time to part time or prefer jobs that provide more flexible work schedules.

e.      Intensity of caregiving: For some individuals caregiving can be more intense or stressful if the care recipient is significantly impaired.

f.        primary caregiver or secondary caregiver: Is the individual included in the study is the primary caregiver? Or another family member in the household is the primary caregiver?

For each of the above variables, please explain why it is not included. It is possible that the dataset does not contain these variables.

3.      Sandwiched group includes caregivers that provide both eldercare and childcare (under the age of 18). The cost of childcare is significant for younger children. I am not sure if it is data limitation that the authors defined childcare for dependents under age 18 and NOT young children. Once again, family size especially older children can help with childcaring responsibilities.

4.      Include interaction term of sandwich and weekly hrs of care. This can capture if the effect of weekly hrs of care on healthcare utilization measures is same or different for "eldercare only" group and "sandwiched" group. On the onset, we should find that the interaction term is insignificant. However, if this interaction term is significant and indicates the impact is more for "only eldercare" group, which can suggest the severity of caregiving services provided in this group. It can also suggest that the household dynamics is different in the "eldercare only" group and "sandwiched" group resulting in an interaction term. With regards to the household dynamics, see the list of variables mentioned in point (2) above.

Since the current study focus is on estimating the cost of caregiving on healthcare system, it may be appropriate that this suggestion can be a follow-up study.

Writing Clarity:

1.      Abstract: In the abstract (line 13), the authors use the term "sandwiched". The sandwich group refers to the group of individuals providing both eldercare and childcare. The term "sandwich" is not so common in the literature and hence authors need to rewrite without referring to the term or provide meaning for the term.

2.      Minor edits. There are some minor editing issues in the paper. For example,

a.      in line 409: "We report an mean of 0.27...", please change it to "We report a mean of 0.27..."

b.      in line 425: a period is missing. "previous research from other countries Specifically, our data suggests a mean monthly" change it to "previous research from other countries. Specifically, our data suggests a mean monthly"

Implications of the study:

1.      The authors have done a commendable job at pointing out the economic implications of family members providing caregiving. Additionally, the estimated cost to society as shown in Table 3 is very interesting. Another cost component is the opportunity cost as documented in the economics of caregiving literature. The labor participation of the entire household may change / differ as a result of a family member being a care recipient. It is possible that one family member quits job (or takes a more flexible job) to be the primary caregiver and another family member picks up more hours of work. Therefore, cost estimated in this paper includes only the medical cost and NOT total societal cost (which could be larger). It would benefit the paper to address this in the discussion section.

Overall, it was a joy to read the paper and I commend the authors on a job well done!

Comments on the Quality of English Language

Minor editing issues. See the word file. Please review the manuscript to correct for any minor issues as mentioned in the word document.

Author Response

Comments and Suggestions for Authors

Overall Impression

Reviewer comment: The authors investigate a significant issue by examining healthcare utlization by caregivers in Canada and highlights the factors associated with utilization using a sample collected from 10 large sized organizations (500+ employees) in 2016. To this end, the authors include two groups of caregivers - "eldercare only" and "sandwich" group (those who provide both eldercare and childcare). This is a valuable area of research that has gained a growing interest in several countries due to aging population and increasing demands on healthcare, thus having an impact on healthcare cost, access, and quality. The paper is generally well-written and well-organized, easy to understand, and clear. I would like to offer some suggestions for improvement to strengthen the paper.

Suggestions:

While the premise of the study is sound, there are a few areas in need of written clarity and additional analyses.

Additional analyses:

  1. The study compares two groups of caregivers - "elder care only" and "sandwiched". The study is unable to compare the caregivers to individuals who do not provide any caregiving services. In the data section (line 178), the authors do mention that the study concentrates on the subset of respondents who engage in eldercare responsibilities. It is possible that the dataset does not contain a control group - no caregiving for the purpose of comparison. If this is the case, please provide in the footnote as to why a control group was not included. However, if the dataset does include a group of individuals that provide no caregiving services, then including them in the analyses can strengthen the implications of the study.

Authors response: Thank you for this suggestion. You are correct that were is no control group for comparison, this was largely because the dataset used was from a rather long national survey specifically on caregiving. It was neither feasible nor expected at the time to prompt non-caregivers to respond to the survey. We have now noted and expanded on this in the limitations sections of the paper.

Reviewer comment:

  1. The study focusses on showing the effect of caregiving on health care utilization. The following variables have not been included in the analysis:
  2. underlying health status of the caregiver: If the caregiver has an underlying health condition (unrelated to caregiving), which can affect the healthcare utilization.

Authors response: this is true, our survey did not probe on the caregivers’ personal health condition as our survey at the time was largely focused on work-related outcomes. We note this now as a limitation in the paper.

Reviewer comment:

  1. relationship between caregiver and elderly person that needs care: Studies have shown that most elder caregivers are spouse of the individual, who themselves may have certain health condition.

Authors response: Thank you for this suggestion. We certainly recognize that the relationship between caregivers and their care recipients—particularly in the case of spousal caregivers—is a critical factor in determining caregiving dynamics and therefore caregiver health. That is why the authors selected to examine broader caregiving categories (that is, just dividing caregivers into eldercare and sandwich) to address trends and commonalities in caregiver experiences. The concern we had was that further disaggregation of the data, would result in greater likelihood of a type I error, particularly given the large amount of variables we are already including in our models.  To alleviate any concerns you might have, we went back to the data and noted that perhaps because all of the caregivers in our sample were all employed (and hence somewhat younger in age) virtually all of them were caring for either their own parents or their partner’s parents. 

Reviewer comment:

  1. family size: Specifically, number of adult members in the household. Larger families may be able to share the caregiving responsibilities. In Table 1, we see that average weekly hours of eldercare is lower for sandwiched group. This could be because sandwiched group could have larger family size and thus responsibilities are shared among other members in the household.

Authors response: Another excellent suggestion, however, again, we chose to limit the number of variables included in our model and highlight the most relevant as defined by the literature. We could have looked at adult members in the household, the number of adults who shared caregiving responsibilities with the caregiver respondent, etc.  This was not, however, the focus of our study which was to look at the relationship between taking on the role of caregiver and the caregivers use of the health care system.  We do, however, now suggest in our directions for future research section that researchers undertake research into the factors that may moderate the relationship between engagement in the caregiver role and the caregiver’s use of the healthcare system.

Reviewer comment:

  1. change in job: Prior studies have documented individuals that provide caregiving change their jobs - full time to part time or prefer jobs that provide more flexible work schedules.

Authors response: Indeed, this phenomenon is well documented within the business/management literature but has primarily been linked to organizational outcomes. In fact, we have other papers exploring this phenomenon.  The focus in this paper was, however, on balancing full-time employment and taking on the role of caregiver for an elderly dependent.  We do, however, now suggest in our directions for future research section that researchers undertake research into the factors that may moderate the relationship between engagement in the caregiver role and the caregiver’s use of the healthcare system such as hours spent in employment related activities per week (e.g., full-time versus part time), use of flexible work arrangements etc. 

Reviewer comment:

  1. Intensity of caregiving: For some individuals caregiving can be more intense or stressful if the care recipient is significantly impaired.

Authors response: Exactly, which is why we use weekly hours of caregiving as a proxy to for intensity. Our assumption, which is backed up by research we have done that is published in the Journal of Health and aging, is that hour in caregiving a week increases concomitant with the type of disorder experienced by the care recipient with caring for someone with dementia being particularly stressful.  We did, however, include this in directions for future research.

Reviewer comment:

  1. primary caregiver or secondary caregiver: Is the individual included in the study is the primary caregiver? Or another family member in the household is the primary caregiver?

Authors response: the way we obtained our sample required all respondents to be the primary caregiver of the care recipient.

Reviewer comment: For each of the above variables, please explain why it is not included. It is possible that the dataset does not contain these variables.

  1. Sandwiched group includes caregivers that provide both eldercare and childcare (under the age of 18). The cost of childcare is significant for younger children. I am not sure if it is data limitation that the authors defined childcare for dependents under age 18 and NOT young children. Once again, family size especially older children can help with childcaring responsibilities.

Authors response: Indeed, the costs of childcare for younger children are higher and different than older children. However, limiting our analysis to younger children only does not provide a full picture of the way in which childcare may impact working parents, particularly as childcare needs differ vastly from family to family and culturally. While it may be appropriate in some contexts to limit or expand upon the age of the child(ren) in question, we felt that on a national level, this would be unjustifiably limiting in scope, particularly as our analysis was exploratory in nature.  Thus intentionally defined childcare to include dependents under age 18 to capture a broader view of the caregiving landscape. We also note that although, in theory, older children can help care for elderly dependents, there is no guarantee that they will in fact engage in any caregiving. Research also shows that the stresses of parent change as the child gets older but do not disappear (as any parent of a teenager today will tell you). 

Reviewer comment:

  1. Include interaction term of sandwich and weekly hrs of care. This can capture if the effect of weekly hrs of care on healthcare utilization measures is same or different for "eldercare only" group and "sandwiched" group. On the onset, we should find that the interaction term is insignificant. However, if this interaction term is significant and indicates the impact is more for "only eldercare" group, which can suggest the severity of caregiving services provided in this group. It can also suggest that the household dynamics is different in the "eldercare only" group and "sandwiched" group resulting in an interaction term. With regards to the household dynamics, see the list of variables mentioned in point (2) above.

Since the current study focus is on estimating the cost of caregiving on healthcare system, it may be appropriate that this suggestion can be a follow-up study.

Authors response: Thank you for this suggestion, it is certainly an interesting suggestion. We agree, this analysis would be better suited for a follow up study, especially as our goal was to explore costs and interactions terms muddy the interpretation of costs. We have, therefore, included this in our suggestions to authors for future research. 

Reviewer comment:

Writing Clarity:

  1. Abstract: In the abstract (line 13), the authors use the term "sandwiched". The sandwich group refers to the group of individuals providing both eldercare and childcare. The term "sandwich" is not so common in the literature and hence authors need to rewrite without referring to the term or provide meaning for the term.
  2. Minor edits. There are some minor editing issues in the paper. For example,
  3. in line 409: "We report an mean of 0.27...",please change it to "We report a mean of 0.27..."
  4. in line 425: a period is missing. "previous research from other countries Specifically, our data suggests a mean monthly"change it to "previous research from other countries. Specifically, our data suggests a mean monthly"

Authors response: Thank you for pointing out these editing errors. We have made the corrections in the text. 

Reviewer comment:

Implications of the study:

  1. The authors have done a commendable job at pointing out the economic implications of family members providing caregiving. Additionally, the estimated cost to society as shown in Table 3 is very interesting. Another cost component is the opportunity cost as documented in the economics of caregiving literature. The labor participation of the entire household may change / differ as a result of a family member being a care recipient. It is possible that one family member quits job (or takes a more flexible job) to be the primary caregiver and another family member picks up more hours of work. Therefore, cost estimated in this paper includes only the medical cost and NOT total societal cost (which could be larger). It would benefit the paper to address this in the discussion section.

 Authors response: Thank you for your time and suggestions on our paper. We have expanded the discussion section to highlight the conservative nature of our estimate. We have also replaced the term “societal” with “medical” where it applies.

Overall, it was a joy to read the paper and I commend the authors on a job well done!

Author's response: thank you for your time and contribution to strengthening this paper!

Reviewer 2 Report

Comments and Suggestions for Authors

The paper tackles an interesting topic: health service utilization among caregivers and the characteristics of caregivers that are associated with higher or lower utilization. However, the theoretical framework/literature review and the methodological approach have several flaws. 

Comments on the Authors.

The article is interesting to read but requires further revisions before it can be accepted for publication. Some of the main concerns are

The theoretical framework / literature review needs to be strengthened. The authors have not developed a strong argument or justification for why the caregiver’s background needs to be studied. The Andersen behavioural model of health services utilization has been widely used to understand the use of services. Perhaps the authors can look at the article “A description of theoretical models for health service utilization: a scoping review of the literature” by Gliedt, J. A., Spector, A. L., Schneider, M. J., Williams, J., & Young, S. (2023) in INQUIRY: The Journal of Health Care Organization, Provision, and Financing, 60(1) to identify different theoretical approaches.

 In the literature, several old references are used to describe estimation. For example, on line 117 it says  “It has been estimated that informal caregivers save the Canadian government $25 billion dollars annually through their unpaid labor (Hollander at el., 2009)”. What is the most recent estimation? I recommend that the authors pay attention to the references. Another example is on line 249: “To extrapolate total societal costs, we assumed a total of 6.1 million caregiver employees in Canada, based on most recent estimates from Statistics Canada (Sinha, 2013)”. Is this reference used because the data was collected in 2016?

The literature does not provide a clear rationale for studying these four services: physician visits, inpatient days at the hospital, emergency room visits, and mental health professional visits. Additionally, including a footnote below Table 1 for the eldercare-only and sandwiched group is not sufficient; this needs to be discussed in the literature.

 Other details are Line 123: A more detailed discussion on “2.3. Government Initiatives.” section is necessary. For example, how permanent residency in Canada for foreign workers can reduce healthcare spending and how various tax credits and employment insurance benefits can reduce caregiving expenses. Line 134-136: What kind of resources? please elaborate.

 Method:

The data used for this paper is from 2016, which is somewhat outdated. Therefore, it’s necessary to justify whether this data is still applicable in the current scenario. It would be beneficial to explore how inflation and other related economic factors have impacted the caregivers. This could be included in the method or the discussion. There is a lot of missing information. For instance, the total sample population in the ten organizations, and the breakdown of public, private, and non-profit organizations within that group. It’s also important to know the response rate and the geographic diversity of these organizations to assess their representativeness. More details are needed.

 It’s also important to justify the “Knowledge worker” choice as an independent variable and explain why this group is particularly important. Furthermore, it’s crucial to consider why the caregivers’ health-related variables are not included as independent or control variables in the analysis, especially when examining the use of healthcare services.

 Estimation for calculation: Why are there different reference years when calculating the estimation?

Line 249: For social cost: “To extrapolate total societal costs, we assumed a total of 6.1 million caregiver employees in Canada, based on most recent estimates from Statistics Canada (Sinha, 2013).”

 Mean cost Line 235: “First, the mean cost to the Canadian healthcare system associated with a single physician visit (across all specialities) is estimated at $73.45 per visit, assuming one service per visit (CIHI, 2022)”.

I would not recommend using the term “sex”; “gender” is more suitable.

Include a statement regarding ethical approval or other ethical considerations related to the survey in the method

 Research question: Authors need to reframe the research question. I assume that “rate” refers to the cost of accessing health services. Why haven’t the authors mentioned two groups of caregivers: eldercare-only and sandwiched in the research question? Also, with the different service groups (i.e., Physician visits, Inpatient days spent at the hospital, Emergency room visits, and Mental Health professional visits)?

Comments on the Quality of English Language

It is easy to read the paper, but there are some inconsistencies in few places. Otherwise, there is nothing to comment on regarding the language.

Author Response

Comments on the Authors.

The article is interesting to read but requires further revisions before it can be accepted for publication. Some of the main concerns are

  1. Reviewer comment: The theoretical framework / literature review needs to be strengthened. The authors have not developed a strong argument or justification for why the caregiver’s background needs to be studied. The Andersen behavioural model of health services utilization has been widely used to understand the use of services. Perhaps the authors can look at the article “A description of theoretical models for health service utilization: a scoping review of the literature” by Gliedt, J. A., Spector, A. L., Schneider, M. J., Williams, J., & Young, S. (2023) in INQUIRY: The Journal of Health Care Organization, Provision, and Financing, 60(1) to identify different theoretical approaches.

Authors response: Thank you for this suggestion. We have now added a section under “caregiver health” where we utilized Pearlin’s stress process model to expand on the theoretical background of poor caregiver health. We also  reference Orstein’s model of healthcare utilization of caregivers of palliative patients.

  1. Reviewer comment: In the literature, several old references are used to describe estimation. For example, on line 117 it says  “It has been estimated that informal caregivers save the Canadian government $25 billion dollars annually through their unpaid labor (Hollander at el., 2009)”. What is the most recent estimation? I recommend that the authors pay attention to the references. Another example is on line 249: “To extrapolate total societal costs, we assumed a total of 6.1 million caregiver employees in Canada, based on most recent estimates from Statistics Canada (Sinha, 2013)”. Is this reference used because the data was collected in 2016?

Authors response: These estimates are provided because they are the latest data available. Since the 2012 General Social Survey, StatsCan has not released data on caregiver employees. As a result of this, few researchers have attempted studies to monetize costs. To address this very valid concern we note this in the section outlining the limitations of this study. 

  1. Reviewer comment: The literature does not provide a clear rationale for studying these four services: physician visits, inpatient days at the hospital, emergency room visits, and mental health professional visits. Additionally, including a footnote below Table 1 for the eldercare-only and sandwiched group is not sufficient; this needs to be discussed in the literature. 

Authors response: We have now added additional information in the methods section on how and why we selected these four services. In addition, we now describe the rationale for studying sandwich caregivers in the literature review section

  1. Reviewer comment: Other details are Line 123: A more detailed discussion on “3. Government Initiatives.” section is necessary. For example, how permanent residency in Canada for foreign workers can reduce healthcare spending and how various tax credits and employment insurance benefits can reduce caregiving expenses. Line 134-136: What kind of resources? please elaborate.

Authors response: we have since elaborated in the government initiatives section.

Method: 

  1. Reviewer comment: The data used for this paper is from 2018, which is somewhat outdated. Therefore, it’s necessary to justify whether this data is still applicable in the current scenario. It would be beneficial to explore how inflation and other related economic factors have impacted the caregivers. This could be included in the method or the discussion.

Authors response:  First, our apologies – the data were collected in 2018. We have corrected this in the manuscript. Second, you are right.  The costs might be much higher now due to inflation. We have put the fact that the data were collected in 2018 as a possible limitation of the study and have noted that in our conclusion that our estimates are conservation and likely underestimate the costs caregiving impose on the health care system. We also suggested additional research in this area to allow researchers to measure the impact of inflation on these costs.

  1. Reviewer comment: There is a lot of missing information. For instance, the total sample population in the ten organizations, and the breakdown of public, private, and non-profit organizations within that group. It’s also important to know the response rate and the geographic diversity of these organizations to assess their representativeness. More details are needed. 

Author’s response: Analysis on an organization-wide level was not conducted because the way that we sampled within these organizations made this impossible. None of the employers who agreed to participate in the study keep information on who within their workforce engaged in caregiving. To get a sample of employed caregivers the employer had, therefore, to send the survey link to all employees with a cover letter indicating that the focus of this survey was on balancing work and caregiving.  To ensure that this was, in fact the case, on the first page of the survey we asked the employee three screening questions:  hours worked to ensure that all respondents were full-time employees, weekly participation in childcare (yes/no) and weekly participation in eldercare (yes/no).  This is mentioned on page five of the manuscript.  While this sampling strategy resulted in an excellent sample of full-time workers with childcare and/or eldercare responsibilities, it did not give us the type of data we needed to calculate response rate per organization.

  1. Reviewer comment: It’s also important to justify the “Knowledge worker” choice as an independent variable and explain why this group is particularly important. Furthermore, it’s crucial to consider why the caregivers’ health-related variables are not included as independent or control variables in the analysis, especially when examining the use of healthcare services.

Authors response: We responded to this query by providing additional information addressing this issue in the measures section of the paper.

  1. Reviewer comment: Estimation for calculation: Why are there different reference years when calculating the estimation? 

Authors response.  Information answering this question was included in the original version of the paper.

  • On Line 249 of the submitted paper we sate: For social cost: “To extrapolate total societal costs, we assumed a total of 6.1 million caregiver employees in Canada, based on most recent estimates from Statistics Canada (Sinha, 2013).
  • In terms of mean cost on Line 235 of the submitted paper we state: “First, the mean cost to the Canadian healthcare system associated with a single physician visit (across all specialities) is estimated at $73.45 per visit, assuming one service per visit (CIHI, 2022)”.

Also relevant is the fact that we use the most recent year available for our various calculations, as needed data was not available for all years.

  1. Reviewer comment: I would not recommend using the term “sex”; “gender” is more suitable. 

Authors response: When we designed the survey, our university ethics board recommended we use the term “sex” rather than gender.  We do not think it appropriate to use the term gender in the paper when the survey collected data on sex.

  1. Reviewer comment: Include a statement regarding ethical approval or other ethical considerations related to the survey in the method.

Authors response: We have included a sentence in the data collection section indicating the steps taken to protect respondent identity and the obtainment of consent from survey respondents.

  1. Research question: Authors need to reframe the research question. I assume that “rate” refers to the cost of accessing health services. Why haven’t the authors mentioned two groups of caregivers: eldercare-only and sandwiched in the research question? Also, with the different service groups (i.e., Physician visits, Inpatient days spent at the hospital, Emergency room visits, and Mental Health professional visits)

Authors response: In the original version of the paper the two groups of caregivers (eldercare and sandwich) were alluded to in our second research question: what characteristics of caregivers are associated with higher or lower utilization?  In response to this question, we have changed the research question so that it now explicitly mentions sandwich vs eldercare caregivers. Furthermore, we have now also stated the service groups used in our analysis.

Reviewer 3 Report

Comments and Suggestions for Authors

This manuscript has a main problem: the sample analyzed was obtained from 10 organizations employing 500+ workers. A subset of 1674 respondents is used in this analysis, but the whole process of selection of companies is not reported and represents a serious limitation for the valid representativeness of the data here reported. Even when the process of how the 1674 is explained the bias on how they were recruited unvalidate the identified estimates.

Author Response

Reviewer's comment: This manuscript has a main problem: the sample analyzed was obtained from 10 organizations employing 500+ workers. A subset of 1674 respondents is used in this analysis, but the whole process of selection of companies is not reported and represents a serious limitation for the valid representativeness of the data here reported. Even when the process of how the 1674 is explained the bias on how they were recruited validate the identified estimates.

Author’s Response:  We have responded to this concern (which was also alluded to by reviewer two) but giving more details on how the sample was selected.  In Canada employers do not collect information on the family demands of their employees (i.e. childcare, eldercare). This made it impossible for us to send out a survey to those with these types of responsibilities only.  We brainstormed how to get an appropriate sample, and the method we came up with (send survey to all employees, use screening questions to ensure that respondents work full time and engage in childcare and/or eldercare activities on a weekly basis) is, in our opinion, defensible given the constraints researchers face in obtaining data from employed caregivers to support the type of analysis undertaken in this paper. We have however now included this as a limitation of our research.  

Round 2

Reviewer 3 Report

Comments and Suggestions for Authors

The authors recognize the limitations of the paper clearly.